# Endobronchial Ultrasound Staging During Navigation Bronchoscopy for Peripheral Pulmonary Nodules in the Real World: Which Patients Will Benefit?

**DOI:** 10.3390/cancers17101700

**Published:** 2025-05-19

**Authors:** Desi K. M. ter Woerds, Roel L. J. Verhoeven, Ad F. T. M. Verhagen, Erik H. J. G. Aarntzen, Erik H. F. M. van der Heijden

**Affiliations:** 1Department of Pulmonary Diseases, Radboud University Medical Center, 6525 GA Nijmegen, The Netherlands; desi.terwoerds@radboudumc.nl (D.K.M.t.W.); erik.vanderheijden@radboudumc.nl (E.H.F.M.v.d.H.); 2Department of Thoracic Surgery, Radboud University Medical Center, 6525 GA Nijmegen, The Netherlands; ad.verhagen@radboudumc.nl; 3Department of Medical Imaging, Radboud University Medical Center, 6525 GA Nijmegen, The Netherlands; erik.aarntzen@radboudumc.nl; 4Department Nuclear Medicine and Molecular Imaging, University Medical Center Groningen, 9713 GZ Groningen, The Netherlands; 5Department of Nuclear Medicine, Eberhard Karls University Tuebingen, 72074 Tuebingen, Germany

**Keywords:** navigation bronchoscopy, peripheral pulmonary nodules, lymph nodes, diagnostic imaging, endobronchial ultrasound, [^18^F]FDG-PET, chest-CT

## Abstract

This single-center study evaluated patients referred for navigation bronchoscopy to diagnose incidentally detected peripheral pulmonary nodules. We focused on the role of endobronchial ultrasound (EBUS) in mediastinal staging within this cohort. Given the size (median, 17 mm) of the detected nodules, systematic EBUS to assess the lymph nodes for possible metastatic disease is not mandatory. We evaluated whether it could be selectively applied in this specific cohort. Our results indicate that EBUS only found metastatic disease in patients with imaging-positive lymph nodes based on Positron Emission Tomography (PET) or contrast-enhanced Computed Tomography (ceCT) reports. No metastases were found in patients with imaging-negative lymph nodes. Performing EBUS only in the cohort of patients with imaging-positive lymph nodes would reduce the number of EBUS procedures by 65.8% without compromising staging accuracy. This selective approach may help optimize lung cancer staging and improve the efficiency of the diagnostic workflow during navigation bronchoscopy.

## 1. Introduction

As lung cancer screening programs are more widely implemented, there has also been an increase in the detection of incidental pulmonary nodules [1]. These nodules are initially evaluated using lung cancer risk prediction scores based on clinical assessment, radiological features, and nodule growth [2]. The increase in pulmonary nodules detected also translates into more nodules being classified as high-risk and this has led to a rise in referrals for minimally invasive diagnostic navigation bronchoscopy (NB) procedures [2,3]. In this population, the reported prevalence of lung cancer in patients referred for NB is high with 76% in our own center and others [4,5].

NB allows for minimally invasive tissue sampling of peripheral pulmonary nodules that are beyond the reach of conventional bronchoscopy, and it offers a favorable safety profile with a lower risk of complications such as pneumothorax [6,7,8]. A frequently cited advantage of NB is the possibility of combining the procedure with immediate staging by endobronchial ultrasound (EBUS) since lymph node staging is equally crucial to determine lung cancer treatment and prognosis. This has become even more important with the recent approval of neoadjuvant (chemo)immunotherapy for patients with stage II and resectable stage IIIa [9,10]. However, the vast majority of patients referred for NB have small (less than 3 cm), peripherally located nodules (ACCP, group D) for which guidelines state that staging is not routinely needed in absence of suspected lymph nodes based on 2′-[^18^F]fluoro-2′-deoxyglucose positron emission tomography ([^18^F]FDG-PET) or computed tomography (CT) imaging [11,12,13,14]. Moreover, patients with a high risk of mediastinal metastases are usually scheduled for upfront EBUS and are not referred for NB.

Systematic EBUS has emerged as the preferred technique for locoregional lymph node staging, replacing more invasive surgical staging procedures [11]. Although guidelines on the performance of EBUS do not typically apply to patients referred for NB, the importance of thorough staging and the fact that NB requires general anesthesia have led to the increased practice of combining EBUS with NB in patients with suspected lung cancer [15,16,17].

However, despite guideline-concordant staging, studies show that a significant number of patients staged as clinical N0 based on CT and/or [^18^F]FDG-PET imaging or EBUS, are upstaged to pN1 or pN2 after surgery [11]. The rate of occult N1 disease can reach up to 16.6% and upstaging to N2 disease occurs in up to 9.9% highlighting the limitations of current staging techniques [11,13,15,18,19].

In this study, we aim to assess the added value of performing systematic EBUS during NB in patients with peripheral nodules. By calculating the number needed to treat (NNT) and evaluating whether certain imaging features, such as [^18^F]FDG-PET and CT findings, can reliably select patients in whom EBUS staging may be omitted, we hope to personalize and refine the diagnostic and staging workflow for this patient population.

## 2. Materials and Methods

This study analyzes clinical and procedural data as prospectively acquired during routine clinical practice in patients who underwent a diagnostic NB between December 2017 and January 2023 in the Radboud University Medical Center, Nijmegen, the Netherlands. During the inclusion period, the Radboud University Medical Center was the only center in the Netherlands performing navigation bronchoscopies and, therefore, evaluated patients from multiple referring centers across the Netherlands. A systematic EBUS was routinely performed in patients scheduled for NB. All patients provided written informed consent to use their data (Reference No. 2017-3706, 2017-3707 and 2019-5148). Patients who underwent an EBUS procedure prior to referral for NB, were excluded. Patients were further excluded from analysis when no NB had been performed based on new clinical findings like an unanticipated central lesion, resorption of the target nodule, or when NB was performed for local treatment purposes. In patients who received a second NB, only the first procedure was analyzed. Patients with a nodule that had an uncertain benign outcome with insufficient or incomplete clinical follow-up of less than 12 months were excluded from analysis, since they could not be categorized correctly, as per international consensus [20,21]. In our center, the NB procedure is based on cone-beam CT-based 3D-image guidance as has been described in detail by Beyaz et al. (2025) [22].

### 2.1. Data Collection and Case Selection Criteria

Patient demographics and nodule characteristics were collected through investigation of electronic medical records, including size, lobe, radiographic characteristics of the nodule, whether the nodule was <2 cm from the pleura and pre-procedural and post-operative TNM-stage (8th edition [23]). Follow-up data from 32 referring hospitals were collected. When data about the specific histology or type of treatment a patient received for lung cancer were inconclusive, lacked detail, or were not available during follow-up from referring hospitals, this was categorized as unknown.

### 2.2. Patient Selection and Imaging Criteria for EBUS Performance Analysis

When an [^18^F]FDG-PET scan was performed up to 6 weeks prior to the NB, [^18^F]FDG-PET outcome of the lymph nodes was scored imaging positive (iN+) when the report of the Nuclear Physician mentioned any uptake of [^18^F]FDG in one or multiple (even bilateral) lymph nodes. When no recent [^18^F]FDG-PET scan was available, lymph nodes were scored on the pre-procedural contrast-enhanced chest CT scan (ceCT) if performed within 6 weeks prior to the NB. The CT outcome was scored iN+ when ipsilateral or mediastinal lymph node(s) with a short axis ≥ 10 mm were found [24]. If despite a negative [^18^F]FDG-PET-outcome the lymph nodes were still suspected to be cN+ based on size on the ceCT scan, the positive CT outcome overruled the negative [^18^F]FDG-PET outcome, and the patient was categorized as iN+. No central review of [^18^F]FDG-PET or ceCT scans was performed, and daily clinical reports were used to categorize nodal status.

### 2.3. Diagnostic Procedures

All patients underwent a cone beam CT (CBCT)-guided NB under general anesthesia as described previously [22]. The choice to combine the procedure with a systematic EBUS was at the discretion of the endoscopist. This decision was not strictly defined and, in general, guided by the available clinical information (i.e., suspected lung cancer versus suspected metastatic pulmonary lesions of non-primary lung cancers), imaging information (i.e., suspected lymph node involvement based on imaging), and/or per-procedural feedback of primary lesion by rapid on-site cytopathology evaluation (ROSE), which was available in all cases.[5]

All EBUS procedures were conducted with a flexible ultrasound bronchoscope (EB-1970UK, Pentax Medical, Tokyo, Japan) using a 22G needle (EchoTip, Cook Medical, Bloomington, IN, USA) for transbronchial needle aspiration (TBNA). Following local and international guidelines, EBUS was systemically performed; stations 4L, 4R, 7, 10L, 10R, 11L, 11Ri, and 11Rs were routinely measured along the short axis. We routinely sample every imaging negative LN > 8 mm, and when PET shows any level of avidity, we aspirate > 5 mm, with clear PET avidity. Also, LN < 5 mm are aspirated when technically feasible, and when there is an applicable EUSb approach it was additionally used routinely and sampled when size > 5 mm and the lymph node characteristics along with clinical indication recommended sampling [12,15,18]. Sampling consisted of at least three TBNA passes or until ROSE established that an adequate sample was obtained. Intraprocedural specimen adequacy was defined based on a sufficient presence of lymphocytes in the smears [25].

### 2.4. Data Analysis

Categorical variables were expressed as absolute and relative frequencies, continuous variables were expressed as means and standard deviations, and non-normally distributed data as median and interquartile range (IQR).

The sensitivity, specificity, negative predictive value (NPV), positive predictive value (PPV), overall accuracy, and number needed to treat (NNT) of nodal imaging to detect N-disease with EBUS were calculated using the standard formulas [26]. The NNT was defined as the number of patients that needed to undergo EBUS to avoid one case of N-disease after surgical resection (pN+), irrespective of the involved location (N1, N2, or N3). The NNT was calculated for all patients, and subgroup analyses were performed for lung cancer patients and/or iN+/iN− patients (Appendix A).

## 3. Results

After exclusion of patients based on the defined criteria, 403 patients were included for study analysis (Figure 1). In these patients, 504 nodules with a median diameter of 17 mm were biopsied (85.9% with a nodule sized < 30 mm and 14% had a mass sized ≥30 mm); 72% of the patients had their lesions located at less than 2 cm from the pleura (Table 1). Of these 403 patients, 138 were defined as having any imaging-positive lymph node involvement (iN+) and 265 as imaging negative (iN−), see also Figure 1. Detailed demographic and lesion data are summarized in Table 1. Lung cancer prevalence in this patient cohort was 67.5%.

### 3.1. EBUS, Imaging and Pathological Outcomes in All Patients

Nodal involvement was scored on [^18^F]FDG-PET imaging in 83.9% of patients, while ceCT was used in the remaining 16.1% of patients. Based on current guidelines, a staging EBUS was not indicated in 64.7% of the cases. In our cohort however, systematic EBUS was performed in 327 patients (81.1%) and diagnosed metastatic lymph node disease in 13 patients with lung cancer and 1 patient with metastatic disease from colon cancer. EBUS was not performed in some cases where malignant origin of the nodule was judged unlikely based on ROSE during NB (e.g., granulomatous origin in combination with symmetric PET avidity suggesting inflammation), at the discretion of the endoscopist or due to unanticipated logistical issues. Surgical follow-up was available in 115 out of 272 lung cancer patients (42.3%). In the surgically treated subgroup, follow-up revealed an additional 17 patients with lymph node metastases and 1 patient with pN2 disease with previous EBUS-based cN1 disease. Of these 17 patients, 11 underwent EBUS-TBNA and in 6 patients, TBNA was not performed due to small lymph node sizes (<7.0 mm) or benign lymph node characteristics. Additionally, 1 of 13 EBUS-diagnosed lymph node metastasis was a false positive finding (cN1 down-staged to pN0). The diagnostic outcome of nodules, EBUS and subsequent performed treatments for both the iN+ and iN− group are summarized in Figure 2 and Figure 3, respectively.

Included patients had a median follow-up time of 738 days (range, 28–2125 days). All 85 patients with a nodule classified as benign had a median follow-up time of 903 days (range, 105–1953 days) and was at least 12 months for non-specific benign nodules that had not resorbed on imaging as per international guidelines [20].

Following the analysis as summarized in Table 2, the sensitivity, specificity, PPV, NPV, and overall accuracy of [^18^F]FDG-PET and CT imaging to predict EBUS outcome are 92.9%, 64.5%, 10.5%, 99.5%, and 65.7%, respectively. The number of EBUS procedures needed to find one patient with metastatic lymph nodes in the overall cohort of 403 patients, the NNT, is 25. When ROSE would provide immediate final diagnosis to prove malignant origin of the target lesion during NB, this would allow selecting only lung cancer patients for EBUS staging in the NB session, the NNT is 10. Details of the NNT calculations can be found in the Appendix A and Table A1.

### 3.2. EBUS in the Cohort of Navigation Bronchoscopy Patients with Imaging-Positive Lymph Nodes

In total, 138 out of 403 patients were found to have any imaging-positive lymph node finding (34.2%). EBUS was performed in 124 out of these 138 patients (89.9%) and detected lymph node malignancy in 12 lung cancer patients and in 1 patient with colon cancer. Eight out of twelve lung cancer patients had a nodule < 30 mm (66.7%).

In this iN+ subgroup, 42 patients (30.4%) were treated surgically. Upon resection, an additional 9 patients with pN+ disease were found, totaling 21 patients with lymph node metastases (Figure 2). In 6 of these 9 patients, surgery was performed within 6 weeks after NB and none received neoadjuvant treatment (Appendix B, Table A2). A more detailed analysis of these patients showed that in only one case this nodal involvement was found in a region outside the reach of EBUS.

The NNT for EBUS to find one patient with metastatic lymph nodes in this sub-cohort was 10 patients. If ROSE would provide sufficient evidence to prove malignant cells during NB and this would allow selecting only lung cancer patients for EBUS staging in the NB session, the NNT would become four patients.

### 3.3. EBUS in the Cohort of Navigation Bronchoscopy Patients with Imaging-Negative Lymph Nodes

A total of 265 out of 403 patients had imaging-negative lymph nodes (65.8%). Herein, EBUS was performed in 203 patients (76.6%) and detected one (false positive) metastatic lymph node. In this case, the pathology report stated that few atypical cells were found, resulting in a multidisciplinary team decision of lymph node metastases (cN1). However, upon surgery no metastatic disease was found (pN0) in this or any other lymph node, thus classifying it as a false positive. No lymph node metastases were found by EBUS in the remainder of the cohort. In a sub-cohort of 73 imaging-negative patients with confirmed lung cancer that also received surgery (27.5%), eight patients were, however, diagnosed with occult metastases after surgery (10.9%, Figure 3). In six of these eight patients, surgery was performed within 6 weeks; none received neoadjuvant treatment and in only one case the nodal involvement was found in a region outside the reach of EBUS (Appendix B, Table A2).

### 3.4. Surgically Verified EBUS Performance

Surgery was performed in 115 patients (Figure 2 and Figure 3, Table 1 and Table 3). In eight lung cancer patients that proceeded to surgery without having performed EBUS—imaging negative (six patients) as well as imaging positive (two patients)—no lymph node metastases were found.

Of the 107 lung cancer patients who underwent surgery after systematic EBUS upon NB, 18 patients (16.8%) were upstaged. One patient was upstaged from cN1 after EBUS to pN2 after surgery, and 17 patients were upstaged from cN0 to pN+ (15.9%).

The ultrasound-based size of all 1377 lymph nodes evaluated by EBUS and aspirated by TBNA and the pathological outcome of EBUS and surgery are listed in Table 3. In the patients undergoing surgery, a total of 34 lymph nodes were involved, of which 12 were inaccessible by EBUS (35.3%), see Table 3. Final cytology revealed insufficient sampling in 99 out of 360 sampled lymph nodes (27.5%), but none of these lymph nodes were found to contain metastases after surgery. Detailed analysis on nodal imaging per patient as compared to EBUS and surgical findings can be found in Appendix C, Table A3.

## 4. Discussion

This single-center study evaluated the added value of EBUS in a not further selected, new cohort of patients with peripheral pulmonary nodules, referred for diagnostic navigation bronchoscopy at a tertiary referral center. When following guidelines strictly, a staging EBUS would not have been indicated in 64.7% of the patients based on imaging. Of the 403 included patients who underwent an NB, 327 patients also underwent an EBUS procedure (81.1%) immediately. Only 12 of these patients were diagnosed with lymph node metastases by EBUS, all of whom had positive lymph nodes on [^18^F]FDG-PET or ceCT imaging (iN+) based on routine clinical reporting. No true metastatic lymph nodes were detected by EBUS in patients with negative imaging (iN−). Surgery however detected another 17 patients (34 lymph nodes) with metastatic disease in both iN+ and iN− patients, of which 35.3% of lymph nodes were inaccessible by EBUS or EUSb.

To assess EBUS effectiveness, the NNT was calculated. In the overall dataset of this study where EBUS had been performed after NB, 25 procedures were required to identify 1 patient with lymph node metastases. A patient stratification scenario where EBUS is limited to only those with positive imaging would have reduced the NNT to 10 patients, without missing any metastases. By performing EBUS only in patients with imaging-positive lymph node findings (based on a [^18^F]FDG-PET and/or ceCT scan), 65.8% of EBUS procedures could be omitted without missing metastatic disease that would have been found by EBUS in this cohort.

### 4.1. Accuracy of Imaging to Predict Lymph Node Involvement as Found by EBUS and/or Surgery

In this study, lymph nodes were scored imaging positive when any [^18^F]FDG uptake was reported in the lymph nodes in the thorax in the routine clinical report. Other methods often use central reading by highly specialized radiologists or specified definitions like a signal larger than the background or a maximal standardized uptake value (SUV_max_) ≥ 2.5. A large Cochrane meta-analysis by Schmidt-Hansen et al. (2014) found that in normal sized lymph nodes the sensitivity of [^18^F]FDG-PET based on activity higher than background, SUV_max_ ≥ 2.5 or ‘other methods’ was 77.4%, 81.3% and 68.0%, respectively, but that the different studies showed a considerable variance hampering interpretation of [^18^F]FDG-PET accuracy [27]. In our study we relied on the original clinical report only, to maximize the potential of our findings to be generalized to other hospitals. In a meta-analysis by Wu et al. (2013) [28] the diagnostic accuracy for staging by ceCT was investigated, comparing this to [^18^F]FDG-PET/CT. In a subgroup of six studies of 806 patients where lymph nodes were assessed by ceCT they report a sensitivity of 74.0% [28]. We found a similar sensitivity of 72.4% in our subgroup of lung cancer patients with a confirmational EBUS and/or surgery (Appendix C, Table A3). In our study, when we retrospectively selected the subgroup of lung cancer patients, [^18^F]FDG-PET and ceCT imaging had an accuracy of 68.9% in detecting nodal involvement, but more importantly, the NPV was 95.0% (Appendix C, Table A3). We agree with Wu et al. that [^18^F]FDG-PET is most likely more sensitive in detecting lymph node metastases than conventional CT alone.

### 4.2. Accuracy of Imaging to Predict if EBUS Will Detect Lymph Node Involvement

A recently published study by Serra Mitja et al. (2024) determined the performance of EBUS-TBNA for mediastinal staging of centrally located T1N0M0 NSCLC [29]. They report an NPV of 98% for EBUS-TBNA and calculated an NNT of 31 patients for the total cohort and 21 patients when only upper lobe tumors are included. They, however, only included T1 patients, and while they took [^18^F]FDG-PET imaging into account, their scoring did not account for [^18^F]FDG-PET-negative lymph nodes that were enlarged on ceCT imaging. In our cohort, we calculated an NNT of 10 in the cohort of patients with imaging-positive lymph nodes, and in theory even to 4 when primary lung cancer patients can be identified and discriminated malignancy from metastatic disease by ROSE [30]. An important difference to Serra Mitja’s study is that our population has peripherally located lesions. However, when hilar node involvement is proven, this may nowadays translate into applying neo-adjuvant chemo-immunotherapy treatment. We, therefore, advocate using both [^18^F]FDG-PET and ceCT characteristics to select patients for EBUS staging.

We have found that all patients with metastatic disease diagnosed by EBUS also had imaging-positive lymph nodes as found on [^18^F]FDG-PET and/or ceCT imaging (excluding a false positive case). This correlation was also described by Chen et al. (2022), who describe that maximal standardized uptake value (SUV_max_) of the lymph nodes was an independent predictor of higher diagnostic accuracy of ROSE in EBUS-TBNA in patients with NSCLC with malignant mediastinal lymph nodes [31]. The sensitivity of 92.9% and NPV of 99.5% of imaging as seen in the presented data suggest that [^18^F]FDG-PET and ceCT could potentially rule out patients in whom EBUS would not detect metastatic disease. However, as the positive predictive value of imaging in our cohort is relatively low (10.5%), it needs to be emphasized that further (minimally invasive) staging remains required to confirm locoregional metastatic disease.

### 4.3. Strengths and Limitations

While this was a single center cohort study, our center was the only center performing navigation bronchoscopies in the Netherlands during the included time frame. This cohort, therefore, most likely represents a more general, but pre-selected population of patients eligible for navigation procedures. We must emphasize that in the Netherlands we do not (yet) have a lung cancer screening program, so our patients had incidental nodules or were detected during evaluation or follow-up of other primary cancers. This may also explain our finding of almost 20% synchronous malignancies in the nodule pathology.

One of the strengths of our study is that we did not perform a central review of all [^18^F]FDG-PET and CT scans to assess imaging N-status for all patients but relied on routine imaging reports. By doing so, we believe that our analysis provides a real-world outcome, but this could also have added more variability. However, when assessing [^18^F]FDG-PET imaging, we focused solely on the [^18^F]FDG uptake in the lymph nodes without considering or adjusting it based on the uptake in the nodule(s), but the physician performing EBUS did use this information when deciding to proceed or refrain from EBUS immediately following NB. As a result, we cannot investigate a possible correlation between the [^18^F]FDG uptake in the primary lesion and the lymph nodes in our cohort, so, as de Leyn et al. state in their guideline, this relation should be taken into account when evaluating nodal involvement on imaging [12].

Another possible bias concerning the performance of EBUS staging is the fact that the endoscopist is not blinded for both the imaging status and the onsite ROSE results of the nodule(s) and/or lymph node aspirations, which may have influenced the decision to perform EBUS or the decision to continue performing EBUS-TBNA or not [5]. There is a possibility that the metastases found during surgery could have been found by EBUS if the endoscopist would not have taken the imaging into account. Nonetheless, performing an EBUS in imaging-negative patients does not seem to provide upstaging and could in our view be omitted in this cohort.

### 4.4. Guidelines and Unexpected Nodal Disease

Current lung cancer staging guidelines do not recommend routine use of EBUS staging for nodules <30 mm [12,25]. While EBUS can be easily combined with an NB in one procedure, the added time and resources raise the question if EBUS can be omitted in certain cases without compromising staging accuracy. Our cohort of patients referred for a primary NB is preselected to exclude patients with high risk of mediastinal or clear hilar metastatic disease and have a median lesion size of 17 mm (85.9% had a lesion of <30 mm). Apart from one false positive lymph node in the iN− cohort, all 12 patients that were diagnosed with metastatic lung cancer through EBUS had imaging-positive lymph nodes. This implies that [^18^F]FDG-PET and ceCT imaging information of lymph node involvement in this cohort of patients can be used for determining the necessity of an EBUS procedure rather than using lesion size alone, in a navigation bronchoscopy cohort. Performing EBUS in patients with [^18^F]FDG-avid lymph nodes, regardless of size, as can be concluded from the results of this study as well, is still according to the guidelines by the ACCP and ESTS [12,14].

Systematic EBUS and EUSb staging in this preselected group could not detect lymph node involvement in 17 patients diagnosed with lung cancer (7.0%) but were detected upon surgery in both the imaging-positive and image-negative group. This aligns with the reported upstaging rates of 6% to 26.7% as found in other studies [18,32,33,34]. Interestingly, in most of these cases (88%), the involved lymph nodes were theoretically accessible for EBUS and/or EUSb (Table 3). Additionally, the sensitivity of EBUS in this cohort of patients is lower than reported in prior EBUS studies and is only 44.4% when looking at these EBUS accessible lymph nodes and 54.5% when these patients had surgery within 6 weeks after NB (Appendix B, Table A2). This difference can, however, be explained by fact that our cohort is selected to have a low prevalence of nodal disease and cannot be compared to prior studies evaluating EBUS performance in a more general population including many more patients with advanced disease [12,13].

## 5. Conclusions

This single-center study demonstrates that the decision to add systemic EBUS staging to a navigation bronchoscopy procedure in patients with peripheral nodules (median size, 17 mm) can be based on [^18^F]FDG-PET or ceCT imaging of the lymph nodes. In our cohort, 81.1% of patients underwent EBUS, yet lymph node metastases were only detected by EBUS in those with imaging-positive nodes, with no true positive findings in imaging-negative cases. These findings suggest that EBUS could be omitted in patients with recent imaging-negative lymph nodes without missing metastases detectable by EBUS. This would reduce the NNT from 25 to 10 patients.

## Figures and Tables

**Figure 1 cancers-17-01700-f001:**
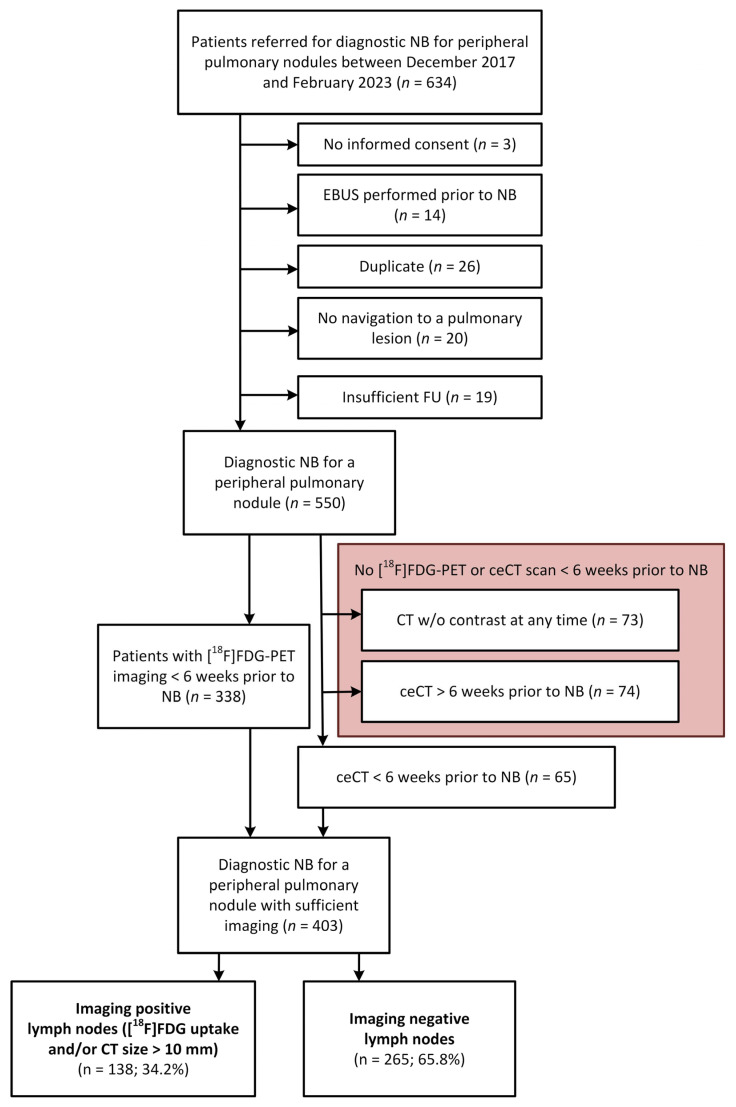
Flowchart of patients receiving a navigation bronchoscopy between December 2017 and February 2023. Abbreviations: ceCT, contrast-enhanced computed tomography; EBUS, endobronchial ultrasound; FU, follow-up; NB, navigation bronchoscopy; [^18^F]FDG, 2′-[^18^F]fluoro-2′-deoxyglucose; PET, positron emission tomography.

**Figure 2 cancers-17-01700-f002:**
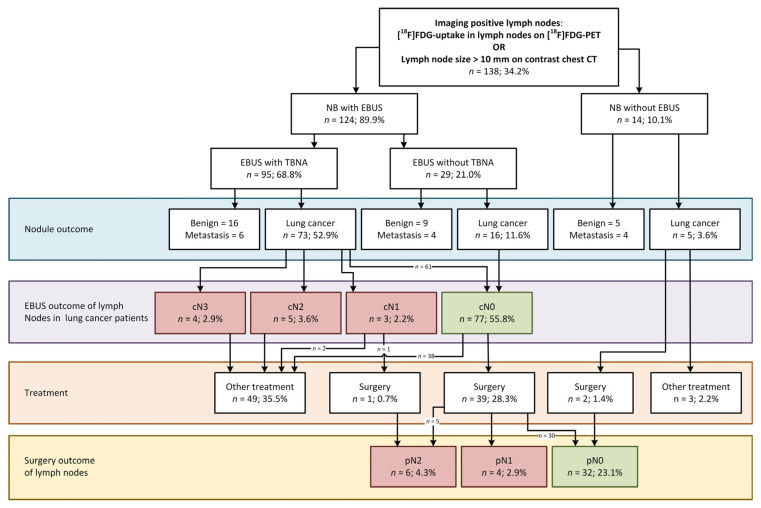
Flowchart of EBUS outcome based on imaging-positive lymph nodes and lesion diagnosis, including N-status after EBUS and/or surgery. Abbreviations: CT, computed tomography; EBUS, endobronchial ultrasound; [^18^F]FDG, 2′-[^18^F]fluoro-‘2-deoxyglucose; FU, follow-up; NB, navigation bronchoscopy; metastasis means pulmonary metastasis of other origin than primary lung cancer; PET, positron emission tomography.

**Figure 3 cancers-17-01700-f003:**
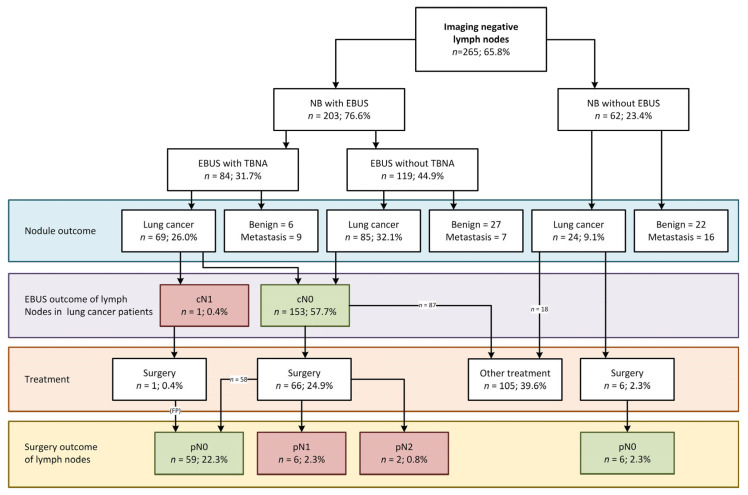
Flowchart of EBUS outcome based on imaging-negative lymph nodes and lesion diagnosis, including N-status after EBUS and/or surgery. Abbreviations: EBUS, endobronchial ultrasound; FP, false positive; NB, navigation bronchoscopy; metastasis means pulmonary metastasis of other origin than primary lung cancer.

**Table 1 cancers-17-01700-t001:** Demographic data of all included patients (*n* = 403) and nodule characteristics and pathological outcome for the sub-selection of patients with a diagnosis of lung cancer (*n* = 272). * The [^18^F]FDG-PET-outcome was scored as positive when any uptake of FDG in one or multiple (even bilateral) lymph nodes was reported in the clinical PET/CT report. Abbreviations: AAH, atypical adenomatous hyperplasia; AC, adenocarcinoma; AIS, adenocarcinoma in situ; BMI, body mass index; CT, computed tomography; EBUS, endobronchial ultrasound; [^18^F]FDG, 2′-[^18^F]fluoro-2′-deoxyglucose positron emission tomography; IQR, interquartile range; MIA, minimally invasive adenocarcinoma; NB, navigation bronchoscopy; NEC, neuroendocrine carcinomas; NSCLC, non-small cell lung cancer; NOS, not otherwise specified; SABR, stereotactic ablative radiotherapy; SCC, squamous cell carcinoma; TBNA, transbronchial needle aspiration.

Patient Baseline Characteristics, All Included Patients (*n* = 403)			Frequency
Patient characteristics	Age, years, median (±IQR)		67 (±11)
	Sex, *n* (%)	Male	205 (50.9%)
		Female	198 (49.1%)
	BMI, median (±IQR)		24.61 (±5.98)
	Lesion < 2 cm from pleura, *n* (%)	Yes	298 (72.2%)
		No	105 (27.8%)
	Radiologic nodule characteristics, *n* (%)	Solid	317 (78.6%)
		Part-solid	47 (11.7%)
		GGO	19 (4.7%)
		Cystic	20 (5.0%)
	Nodule size, mm on CT, median (±IQR)		17 (±12)
	Total number of nodules navigated to, *n*		504
	Number of nodules evaluated per patient, *n* (%)	One nodule	308 (76.4%)
		Two nodules	89 (22.1%)
		Three nodules	6 (1.5%)
Imaging data	[^18^F]FDG PET availability < 6 weeks prior to NB, *n* (%)		338 (83.9%)
	Time between [^18^F]FDG-PET and NB, days, median (±IQR)		22 days (±14)
	[^18^F]FDG-uptake in lymph nodes, *n* (%)	Positive *	122 (36.1%)
		Negative	216 (63.9%)
	In patients without PET or with PET > 6 weeks prior to NB: CT with contrast only < 6 weeks prior to NB, *n* (%)		65 (16.1%)
	Time between CT contrast and NB, days, median (±IQR)		24 days (±15)
	CT outcome of lymph nodes, *n* (%)	Positive	19 (29.2%)
		Negative	46 (70.8%)
EBUS	EBUS performed, *n* (%)	Yes	327 (81.1%)
		No	76 (18.9%)
	TBNA performed, *n* (%)	Yes	179 (54.7%)
		No	148 (45.3%)
	Number of lymph nodes sampled, median (±IQR)	Imaging-positive group	2 (±2)
		Imaging-negative group	1 (±1)
**Subgroup analysis in patients with a final diagnosis of lung cancer (*n* = 272)**			**Frequency**
Pre-procedural characteristics	CT stage, *n* (%)	Tis	5 (1.8%)
		T1a	39 (14.3%)
		T1b	100 (36.8%)
		T1c	46 (16.9%)
		T2a	18 (6.6%)
		T2b	12 (4.4%)
		T3	15 (5.5%)
		T4	4 (1.5%)
		Multiple lesions (all T)	33 (12.1%)
	iN-stage (based on PET and or contrast CT imaging), *n* (%)	iN−	178 (65.4%)
		iN+	94 (34.6%)
	Pre-NB/-EBUS imaging-based cN-stage of all iN+ patients, *n* (%)	cN0	56 (59.6%)
		cN1	18 (19.1%)
		cN2	15 (16.0%)
		cN3	5 (5.3%)
	Lesion location, *n* (%)	Right upper lobe	98 (36.0%)
		Right middle lobe	10 (3.7%)
		Right lower lobe	38 (14.0%)
		Left upper lobe	60 (22.1%)
		Left lower lobe	29 (10.7%)
		Multiple lobes	37 (13.6%)
Post-procedural characteristics	Post-NB+/-EBUS cN-stage, *n* (%)	cN0	259 (95.2%)
		cN1	4 (1.5%)
		cN2	5 (1.8%)
		cN3	4 (1.5%)
	Treatment, *n* (%)	Surgery	115 (42.3%)
		SABR and/or cyberknife	105 (38.6%)
		Combination of treatments	29 (10.7%)
		Best supportive care	15 (5.5%)
		Immunotherapy	4 (1.4%)
		Chemotherapy	3 (1.1%)
		Unknown	1 (0.4%)
	Neo-adjuvant treatment, *n* (%)	Yes	10 (8.7%)
		No	105 (91.3%)
	pN-stage after surgery, *n* (%)	pN0	97 (84.3%)
		pN1	10 (10.3%)
		pN2	8 (8.2%)
	Tumor histology, *n* (%)	AC (incl. AAH, ACIS, and MIA)	138 (50.7%)
		SCC	38 (14.0%)
		Unknown (no tissue)	21 (7.7%)
		Carcinoid	8 (2.9%)
		NSCLC(-NOS)	4 (1.5%)
		SCLC	8 (2.9%)
		Mixed AC-SCC	2 (0.7%)
		Multiple (synchronous) histologically different lung cancers	53 (19.5%)

**Table 2 cancers-17-01700-t002:** The sensitivity, specificity, PPV, and NPV of PET/CT imaging in the cohort of patients who received a navigation bronchoscopy and EBUS, irrespective of pathology outcome. ^1^ Twelve lung cancer patients and one lymph nodal involvement of metastatic disease from colon cancer. ^2^ One false-positive lymph node. Abbreviations: EBUS, endobronchial ultrasound; NB, navigation bronchoscopy; NNT, number needed to treat; NPV, negative predictive value; PPV, Positive Predictive value.

Accuracy of Nodal Imaging in Navigation Bronchoscopy Patients			
All NB + EBUS patients (*n* = 327)	EBUS outcome		
	cN+	cN-	Total
iN+	13 ^1^	111	124
iN−	1 ^2^	202	203
Total	14	313	327
Sensitivity	13/(13 + 1)		92.9%
Specificity	202/(111 + 202)		64.5%
PPV	13/(13 + 111)		10.5%
NPV	202/(1 + 202)		99.5%
Overall accuracy	(13 + 202)/327		65.7%
NNT overall			25
NNT iN+ subgroup			10
NNT iN− subgroup			∞

**Table 3 cancers-17-01700-t003:** EBUS and surgery outcomes: average lymph node size per region, number of times a lymph node was sampled and the pathological outcome per lymph node station after EBUS and after surgery in all patients referred for NB. In multiple patients, more than 1 lymph node was found to contain metastatic cells. * Station 11Ri and 11Rs are separately indicated in EBUS reports, but not separately evaluated in the pathology report of surgically removed tissue.

Lymph Node Station (#)	2L	4L	10L	11L	≥12L	3	5	6	7	8	9	2R	4R	10R	11Ri	11Rs	≥12R	Total
Number of times reported and measured, *n*	7	239	12	218	0	0	0	0	253	0	0	14	235	53	145	201	0	1377
Average short axis on EBUS (mm)	3.60	4.40	5.05	5.35	-	-	-	-	6.20	-	-	4.90	4.95	4.60	5.30	5.40	-	5.10
Number of times sampled, *n*	0	41	2	64					83			5	59	6	29	71		360
Benign, *n*	0	20	2	41					66			3	41	2	22	45		242
Insufficient specimen, *n*	0	16	0	22					16			2	13	3	6	21		99
Malignant, *n*	0	5	0	1					1			0	5	1	1	5		19
Malignant after surgery (pN+), *n*	0	1	0	3	5	0	0	1	3	1	0	1	4	4	5 *		6	34

## Data Availability

The data that support the findings of this study are available from the corresponding author upon reasonable request.

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
