# Peer review of "Endobronchial Ultrasound Staging During Navigation Bronchoscopy for Peripheral Pulmonary Nodules in the Real World: Which Patients Will Benefit?"

_cancers, 2025, doi:10.3390/cancers17101700_

Round 1
Reviewer 1 Report
Comments and Suggestions for Authors
This is an interesting and well done study evaluating the role of EBUS-TBNA for hilar-mediastinal staging in patients referred for small peripheral pulmonary nodules.
403 patients were included in the analysis.
After a careful and well conducted evaluation, the authors conclude that EBUS could be omitted in patients with imaging-negative lymph node.
The study is interesting, methodologically well conducted and it may have a useful impact in clinical practice.
I have no major comments.
Just a minor comment: line 129 is repeated twice in line 130.
Author Response
This is an interesting and well done study evaluating the role of EBUS-TBNA for hilar-mediastinal staging in patients referred for small peripheral pulmonary nodules. 403 patients were included in the analysis. After a careful and well conducted evaluation, the authors conclude that EBUS could be omitted in patients with imaging-negative lymph node. The study is interesting, methodologically well conducted and it may have a useful impact in clinical practice. I have no major comments. Just a minor comment: line 129 is repeated twice in line 130.
We thank the reviewer for the positive and encouraging feedback on our manuscript. We appreciate your recognition of the methodological quality and potential clinical impact of our study.
Regarding your minor comment: thank you for pointing out the repetition in lines 129 and 130. We have removed the duplicate sentence.
Reviewer 2 Report
Comments and Suggestions for Authors
In this single-center study, the authors reviewed their EBUS/NAV data for 403 patients with mainly peripheral lung nodules with an average diameter of 17mm. EBUS was done in 81.1% of the patients. When EBUS was done based on radiographic lymphadenopathy, which is defined as enlarged by more than 10mm or PET positive, malignant lymph nodes were found in 8.4%(12 patients) by EBUS. 5.4% (9 patients) were missed by EBUS and diagnosed subsequently by surgery. In the group of patients with radiographically negative lymph nodes, no metastatic lymph nodes were found by EBUS, but surgery revealed occult nodal metastasis in 3.1%(8 patients). The author concluded that EBUS may be safely omitted in patients referred for navigational bronchoscopy when no pathological lymphadenopathy is found on PET scan or CT chest with contrast.
The study is well written with clear analysis, informative Figures and tables. The conclusion supports current guidelines recommending EBUS for lung nodules >3cm, central tumor and cN1 disease. The study describes an important issue related to the appropriate use of EBUS in combination with navigational bronchoscopy.
This is a single-center retrospective study to evaluate the role of EBUS in staging the mediastinum. The lack of standardized protocol to define which patient undergoes an EBUS, the minimum size of the lymph node to sample, and the number of stations to sample, combined with the clinical knowledge of the results of the CT and PET scan may have affected the pretest probability of malignancy in the Lymph nodes and accuracy of the EBUS Staging. I recommend minor revisions as below:
a- There is no protocol defined for which patients get an EBUS, which is left to the discretion of the physician. Please add to the limitation
b- Please describe the institution's protocol regarding the lymph node size to sample during EBUS. In general, any Lymph node more than 5 mm is sampled to provide complete staging.
c- Please add the number of lymph node stations sampled per patient in both groups (iLN+ and iLN-).
d- Please adjust the conclusion in the abstract to reflect the current population, which includes patients with small peripheral lymph nodes with radiographically negative lymph nodes.
1- Line 122: Please indicate if the short axis of the lymph node was used to define abnormal adenopathy
2- Line 129: duplicate- please delete
3- Figure 1: the abstract lists the number of iLN+ as 141 but the figure lists it as 138- please adjust
4- Table A1- the denominator in the NNT calculation is listed as 126+201 which is different than what is listed in Figure 2 and Figure 3 (NB with EBUS 124 and 203 respectively)
Author Response
In this single-center study, the authors reviewed their EBUS/NAV data for 403 patients with mainly peripheral lung nodules with an average diameter of 17mm. EBUS was done in 81.1% of the patients. When EBUS was done based on radiographic lymphadenopathy, which is defined as enlarged by more than 10mm or PET positive, malignant lymph nodes were found in 8.4% (12 patients) by EBUS. 5.4% (9 patients) were missed by EBUS and diagnosed subsequently by surgery. In the group of patients with radiographically negative lymph nodes, no metastatic lymph nodes were found by EBUS, but surgery revealed occult nodal metastasis in 3.1% (8 patients). The author concluded that EBUS may be safely omitted in patients referred for navigational bronchoscopy when no pathological lymphadenopathy is found on PET scan or CT chest with contrast.
The study is well written with clear analysis, informative Figures and tables. The conclusion supports current guidelines recommending EBUS for lung nodules >3cm, central tumor and cN1 disease. The study describes an important issue related to the appropriate use of EBUS in combination with navigational bronchoscopy.
This is a single-center retrospective study to evaluate the role of EBUS in staging the mediastinum. The lack of standardized protocol to define which patient undergoes an EBUS, the minimum size of the lymph node to sample, and the number of stations to sample, combined with the clinical knowledge of the results of the CT and PET scan may have affected the pretest probability of malignancy in the Lymph nodes and accuracy of the EBUS Staging. I recommend minor revisions as below:
a- There is no protocol defined for which patients get an EBUS, which is left to the discretion of the physician. Please add to the limitation.
We would like to thank the reviewer for the thoughtful evaluation and supportive comments regarding the importance and quality of our study.
We agree that, alongside the possible bias in EBUS performance, performing EBUS in general was left up to the endoscopist and have added a statement on this limitation in line 448-450 stating that this decision is up to the physician and therefore not performed in all patients.
b- Please describe the institution's protocol regarding the lymph node size to sample during EBUS. In general, any Lymph node more than 5 mm is sampled to provide complete staging.
We routinely sample every imaging negative LN >8 mm, and when PET shows any level of avidity, we aspirate >5 mm, with clear PET avidly also LN<5 mm are aspirated when technically feasible and when applicable EUSb approach was additionally used routinely. This protocol has been used in our center for many years as published earlier (J Clin Oncol 2023 Vol. 41 Issue 22 Pages 3805-3815; Chest 2015 Vol. 147 Issue 1 Pages 209-215; JAMA 2013 Vol. 309 Issue 23 Pages 2457-2464. These details of this routine have been added in the manuscript line 153-156.
c- Please add the number of lymph node stations sampled per patient in both groups (iLN+ and iLN-).
We added the number of sampled stations in the imaging positive and imaging negative groups in Table 1, 2 and 1 lymph node stations respectively.
d- Please adjust the conclusion in the abstract to reflect the current population, which includes patients with small peripheral lymph nodes with radiographically negative lymph nodes.
We have adjusted the conclusion in line 49 to include patients with peripheral pulmonary nodules.
- Line 122: Please indicate if the short axis of the lymph node was used to define abnormal adenopathy
We added that the short axis was used for measurements of the lymph node in line 134 (previously line 122).
2- Line 129: duplicate- please delete
This line is deleted.
3- Figure 1: the abstract lists the number of iLN+ as 141 but the figure lists it as 138- please adjust
We would like to thank the reviewer for noticing this omission. The correct number is 138. We have adjusted it in the abstract and checked the rest of the manuscript for similar errors.
4- Table A1- the denominator in the NNT calculation is listed as 126+201 which is different than what is listed in Figure 2 and Figure 3 (NB with EBUS 124 and 203 respectively)
This is indeed not correct, another excellent observation by the reviewer; thank you so much. We have adjusted this to report the correct numbers to 124 and 203 as per Figure 2 and 3, the calculations and therewith the NNT were checked and have not changed.
Reviewer 3 Report
Comments and Suggestions for Authors
We read with great interest the manuscript titled “EBUS Staging During Navigation Bronchoscopy for Small Peripheral Pulmonary Nodules in the Real-World: Which Patients will Benefit?” submitted to Cancers by ter Woerds et al. In this prospective case series, the authors describe their experience at a single center performing staging EBUS in patients undergoing navigational bronchoscopy using cone beam CT for diagnosis of pulmonary nodules. Based on the data presented, the authors conclude that systematic EBUS staging can be deferred in patients with negative nodal imaging findings on pre-procedural imaging (FDG-PET-CT or contrast-enhanced CT), as the NNT if the full cohort was calculated as 25. The manuscript is well-written and provides comprehensive detail in a clear and organized manner. The study question is valid and of interest to the lung cancer community.
Major comments:
- The study title and narrative emphasize a focus on the diagnosis and management of small and peripheral pulmonary nodules. A pulmonary nodule is defined as <3 cm in the largest diameter, whereas a lesion >3 cm is defined as a mass. The guidelines regarding the need for mediastinal LN staging in patients with nodules vs. those with masses differ. Similarly, the guidelines regarding the need for EBUS staging in central vs. peripheral lesions differ. While the mean and median diameter of lesions included in this study meet the nodule size criterion, about 14% of lesions in the cohort can be classified as masses. This also contradicts the authors’ definition of “small” peripheral nodule, as nodules and masses of various sizes were included; not all were necessarily “small”. Furthermore, the definition of peripheral location was not clearly delineated in the manuscript and the authors need to clarify whether patients with central lesions were excluded and how centrality was defined. Since the study’s hypothesis and primary aims are to explore the need of mediastinal LN staging in patients with small and peripheral nodules, the authors should reconsider the inclusion criteria of this study in a manner that addresses the study hypothesis with outmost accuracy.
- The study conclusions indicate that mediastinal LN staging by EBUS can be deferred in patients with small peripheral nodules and no evidence of abnormal adenopathy by FDG avidity on PET-CT or on contrast enhanced CT. Two major concerns require the authors attention regarding the methodology of radiographic evaluation: a) The inclusion criteria imposed in the study mandated these imaging studies be performed within 6 weeks of the bronchoscopic intervention. The choice of 6 weeks and the reliance primarily on PET-CT are not aligned with current guidelines. Furthermore, these inclusion criteria limit the generalizability of this study’s results over real-world setting and over health systems in which accessibility to advanced imaging is more limited. This diminishes the strength of the recommendation made in the conclusion of this study. The authors should rationalize the choice of 6 weeks interval and address the limited generalizability of their results. b) Despite being a prospective trial, this study lacked central review of the pre-bronchoscopy imaging and relied on review of reports. While the authors count this as a strength in their discussion, central review of imaging is fundamental to prospective studies, as reliance on reports is biased in many ways and does not allow direct confirmation of the finding and adequate triage of cases into the study workflow. Since radiographic findings are central to this manuscript’s hypothesis, aims, and conclusions, I would expect the authors to address this limitation in a more comprehensive manner.
Methodology:
- Despite being a prospective study, the performance of EBUS was not protocolized and was left to the discretion of the operator based on a clinico-radiographic preoperative assessment as well as intra-operative ROSE-based findings. The reliance on ROSE to guide EBUS staging is debatable and not established in the literature. It was previously established that the specificity and sensitivity of ROSE for cancer is 1) highly variable across institutions, and 2) not considered reliable for intra-operative decision making. Furthermore, individual provider decision regarding the need or lack of for EBUS staging may be biased in many ways. As one example, it introduces a bias into the data by decreasing the potential number of iN-/pN+ cases, which comprises the main conclusion of this manuscript.
- Patients with ceCT performed >6 weeks prior to NB were excluded from the analysis. Please rationalize this exclusion criterion. It is reasonable to assume that the vast majority of patients included in the study had cross-sectional imaging that identified the target lung nodule. The authors should explain: (a) why a patient that had a ceCT >6 weeks ago and has (for example) a lung nodule with a hilar lymph node would be excluded from this study, (b) why 6 weeks (please provide a reference if available), and (c) was the 6-weeks criterion also applied to FDG PET studies.
- Line 164-166 – suggest flipping with the two sentences for better flow. Would be helpful to provide a summary of the diagnostic yield, diagnostic accuracy, and the lung nodule sampling range of pathologic diagnoses. What was the outcome of patients with nondiagnostic peripheral lung biopsy by NB? Along the same line, in Table 1 it is noted that in 21 cases the tumor histology is classified as “unknown” – how was the diagnosis of lung cancer made? This needs to be clarified in the manuscript.
- Table 1- “combination of treatments” – please elaborate. Are neoadjuvant cases included here? Radio-chemotherapy? Also- chemotherapy is now frequently combined with immunotherapy. The separation between the two is unclear. Please provide a more detailed breakdown.
- Figure 1: please expand the greyed out box. The box title state PET or CT but there is no reference to how many patients were excluded due to lack of PET imaging. Also, patients with non-contrast enhanced CT were also excluded regardless of timeframe. This is not reflected in the box title.
- Based on Table 3, it appears that 27% (99/360) of LN samplings were non-diagnostic. There is no commentary about the outcome of these staging cases or how they were calculated in sections 3.2 or 3.3. Furthermore, this rate of non-diagnostic EBUS sampling appears to be higher than that reported on prior literature. The authors should comment on these two concerns.
- As in previous studies, occult nodal disease found on surgical resection has been attributable to lymph nodes not accessible by EBUS. Previous studies have found that EUS-FNA increases the detection of occult lymph node metastases. Please provide some rationale about why this should or should not be considered for similar patients undergoing navigational bronchoscopy, and the logistical or educational challenges that may ensue.
- The discussion section could benefit from a revision. Some of the discussions on how best to utilize PET-CT vs. CT with EBUS-TBNA could be more succinct. Instead of organizing the discussion by concept, would recommend using standard sections:
- Restate research question/hypothesis
- Summarize key findings
- Interpret your findings
- Compare to existing literature
- Provide limitations
- Provide implications
- Provide future directions
Minor comments:
Abstract:
- Prevalence of lung cancer in patients referred for NB is high- based on what data? Several of the lung cancer screening studies, such as NLST, show otherwise.
Introduction:
- Lines 58-60 – the references provided do not support the statement. Please provide substantial references or revise the statement accordingly.
- For lines 58-59 in the introduction: “The growing number of nodules...” recommend rephrasing.
- Lines 60-61 - the references provided do not support the statement. Please provide substantial references or revise the statement accordingly.
- For lines 60-61, recommend removing, “As a result,” and rephrase as “In addition, the prevalence of lung cancer...” Would also recommend providing numbers of lung cancer prevalence in patients getting navigational bronchoscopy from references 4 and 5.
- Line 62 – the term “effective” is vague. Please revise to provide a meaningful statement that truly reflects the role and efficacy of NB in the diagnosis of peripheral lung nodules (which is not as good as TTNB).
- Line 70 – “All patients” - Based on what data? Please provide a supportive reference or revise.
- Line 74, “Although these criteria...” it is unclear what is meant by criteria, please clarify or rephrase.
- Line 79-80 – the deabbreviated FDG should be moved up to line 70
- Line 84: the use of the term “small” is misleading. The study included lesions both smaller and greater than 3 cm, which includes both nodules and masses. Please revise.
Methods:
- Lines 103-105 – this exclusion criterion is unclear. Please redefine “uncertain benign outcome with insufficient or incomplete clinical follow-up” – this statement is unclear. The cited Delphi statement does not indicate that these cases should be excluded, but rather cannot be included in accuracy analyses. If that was the reason for exclusion, please state that clearly in the methodology.
- Line 109 – please redefine “aspect of the nodule”
- Line 112 – data is a plural term. Please correct.
- Remove line 129 in Methods, it is a repeated sentence of line 130.
Results:
- How many cases had ROSE available?
- Line 160 – consider reporting median with range (instead of mean) to align with Table 1.
- Line 188 – the authors should provide quantifiable details about the number of cases where EBUS was not performed.
- Lines 196-197 – recommend moving the statement regarding available surgical resection data earlier in this paragraph for more clarity.
- Although briefly mentioned under sections 3.2 and 3.3, would interesting to include data on the number of neoadjuvant cases included in this study and whether that affected pre- and post-resection staging.
- The results would benefit from adding the median number of lymph nodes sampled per patient in
- Consider including data—if available—on median procedural time for navigational bronchoscopy and when combined with EBUS.
- Table 1, when EBUS-TBNA was performed. This is helpful for future studies.
- Table 1: consider changing “aspect” to “radiographic consistency”
- Table 1: nodules per patient: best to provide a summary of how many cases included 1 targeted lesion, 2 targeted lesions and so forth. Providing a mean or a median (unclear what is reported in the table) is not clear.
- Figure 2 – some numbers do not add up. For example: under “EBUS outcome of lymph nodes in lung cancer patients”, under cN0 there are 63+16 patients, but the n is indicated as 77. The authors should revisit the figure and revise the counts.
Discussion:
- Lines 352 and 281 are contradictory and need to be clarified. In line 352: “Our cohort of patients referred...” describes the patient population as “heavily preselected to exclude patients with a high risk of mediastinal [disease].” In line 281 of the Discussion, the cohort is described as an “unselected group of patients.”
- Lines 352-354, consider rephrasing for ease of reading.
- In reference to lines 357-360 in the discussion, “This implies that...” it should also be stated that the study findings support the current ACCP and ESTS guidelines that mediastinal staging should be considered for FDG-avid lymph nodes regardless of lesion size.
Author Response
We read with great interest the manuscript titled “EBUS Staging During Navigation Bronchoscopy for Small Peripheral Pulmonary Nodules in the Real-World: Which Patients will Benefit?” submitted to Cancers by ter Woerds et al. In this prospective case series, the authors describe their experience at a single center performing staging EBUS in patients undergoing navigational bronchoscopy using cone beam CT for diagnosis of pulmonary nodules. Based on the data presented, the authors conclude that systematic EBUS staging can be deferred in patients with negative nodal imaging findings on pre-procedural imaging (FDG-PET-CT or contrast-enhanced CT), as the NNT if the full cohort was calculated as 25. The manuscript is well-written and provides comprehensive detail in a clear and organized manner. The study question is valid and of interest to the lung cancer community.
Major comments:
- 1-
The study title and narrative emphasize a focus on the diagnosis and management of small and peripheral pulmonary nodules. A pulmonary nodule is defined as <3 cm in the largest diameter, whereas a lesion >3 cm is defined as a mass. The guidelines regarding the need for mediastinal LN staging in patients with nodules vs. those with masses differ. Similarly, the guidelines regarding the need for EBUS staging in central vs. peripheral lesions differ. While the mean and median diameter of lesions included in this study meet the nodule size criterion, about 14% of lesions in the cohort can be classified as masses. This also contradicts the authors’ definition of “small” peripheral nodule, as nodules and masses of various sizes were included; not all were necessarily “small”. Furthermore, the definition of peripheral location was not clearly delineated in the manuscript and the authors need to clarify whether patients with central lesions were excluded and how centrality was defined. Since the study’s hypothesis and primary aims are to explore the need of mediastinal LN staging in patients with small and peripheral nodules, the authors should reconsider the inclusion criteria of this study in a manner that addresses the study hypothesis with outmost accuracy.
We would like to thank the reviewer for the for the very thorough and detailed evaluation of our work, the positive constructive feedback on our manuscript and we the recognition regarding the importance and quality of our study.
We would like to remind the reviewer that this was a detailed analysis of the real-world clinical practice of our tertiary referral center for navigation bronchoscopy. As such, we did not further select patients, but indeed this is a preselected population. As described, all patients with an increased risk of mediastinal or hilar involvement were triaged for an EBUS-first procedures if this was not already done in the referring hospital. Likewise, when a-priori, the chance of diagnostic success of ‘normal’ rEBUS supported bronchoscopy was judged high, patients were not triaged for CBCT-based navigation bronchoscopy.
Indeed, 86% of our cases had nodules and 14% had masses but this size is the effective maximal size in any direction, and despite the larger size, clear clinical need for image guided transbronchial biopsy existed, i.e. these patients were inaccessible for TTNB or standard bronchoscopy. The nodules/masses were indeed peripheral, since 72% of lesions were located within 2 cm of the pleura.
Following your suggestion, and to avoid misinterpretation, we have changed the title of our manuscript and deleted all references to “small” peripheral pulmonary nodules and clarified that 14% can be classified as masses, in line 176-177.
- 2-
The study conclusions indicate that mediastinal LN staging by EBUS can be deferred in patients with small peripheral nodules and no evidence of abnormal adenopathy by FDG avidity on PET-CT or on contrast enhanced CT. Two major concerns require the authors attention regarding the methodology of radiographic evaluation:
a) The inclusion criteria imposed in the study mandated these imaging studies be performed within 6 weeks of the bronchoscopic intervention. The choice of 6 weeks and the reliance primarily on PET-CT are not aligned with current guidelines. Furthermore, these inclusion criteria limit the generalizability of this study’s results over real-world setting and over health systems in which accessibility to advanced imaging is more limited. This diminishes the strength of the recommendation made in the conclusion of this study. The authors should rationalize the choice of 6 weeks interval and address the limited generalizability of their results.
While we agree with the reviewer that advanced [18F]FDG-PET imaging may not be accessible everywhere, it is available in most – if not all – centers that offer advanced navigation bronchoscopy, which forms the basis of our analysis. So while it could diminish the strength of our recommendations regarding the performance of EBUS in other populations, in the context of our study, namely the navigation bronchoscopy population with peripherally located lesions and in the absence of clear mediastinal disease, our analysis is in our view very valuable and relevant.
In regard to PET/CT vs ceCT discussion: nowadays in routine clinical practice, nodule follow up is often done with low-dose CT without contrast. When volume doubling time shows increased risk, most often [18F]FDG-PET imaging is ordered additionally and as a result often ceCT is not repeated and not available at the time of referral for the navigation procedure.
Regarding time interval for imaging, we disagree with the reviewer’s statement that this is a major limitation. In fact we feel that by using this strict cut-off we are able to better relate imaging characteristics to the final pathology outcome. The 6 week cut-off may not be a guideline supported value, but is used in our MDT and many other centers as a maximum time interval between imaging and start of surgical treatment and follows our national guideline in the Netherlands (available in Dutch at this link. However, we have also discussed this cut-off at length before finalizing our analysis and for your information can share that also without this 6 week cut-off, the final conclusions with respect of detection of nodal involvement by EBUS in the imaging positive group and no detection of nodal involvement by EBUS in de imaging negative group remain unchanged. But without a clear value for this timeline, it will be more difficult for other centers to compare their finding with our observations.
-3-
b) Despite being a prospective trial, this study lacked central review of the pre-bronchoscopy imaging and relied on review of reports. While the authors count this as a strength in their discussion, central review of imaging is fundamental to prospective studies, as reliance on reports is biased in many ways and does not allow direct confirmation of the finding and adequate triage of cases into the study workflow. Since radiographic findings are central to this manuscript’s hypothesis, aims, and conclusions, I would expect the authors to address this limitation in a more comprehensive manner.
We thank the reviewer for addressing this important topic. Our data was collected prospectively, but we did not conduct a (randomized) clinical trial. Our study in this specific population aimed to use real-world clinical data and we are convinced that our data is very useful for the community, and as such we also and intendedly refrained from central reviewing CT and PET imaging and used routinely available clinical reports so it would best match the real-world variability and human error in these diagnostics. Following your suggestions, in the revised manuscript we have rephrased the related section in the discussion (line 446).
-4-
Methodology:
- Despite being a prospective study, the performance of EBUS was not protocolized and was left to the discretion of the operator based on a clinico-radiographic preoperative assessment as well as intra-operative ROSE-based findings. The reliance on ROSE to guide EBUS staging is debatable and not established in the literature. It was previously established that the specificity and sensitivity of ROSE for cancer is 1) highly variable across institutions, and 2) not considered reliable for intra-operative decision making. Furthermore, individual provider decision regarding the need or lack of for EBUS staging may be biased in many ways. As one example, it introduces a bias into the data by decreasing the potential number of iN-/pN+ cases, which comprises the main conclusion of this manuscript.
We agree with the reviewer that concordance of ROSE with final results is highly variable between institutions and that the literature on this also shows non-uniform results. As this is real-world data from our highly experienced center, we also are lucky to have a highly dedicated team of cyto-technicians that have been supporting us for over 20 years. In our center we can rely on their accuracy.
In our center, we routinely sample every imaging negative LN >8 mm, and when PET shows any avidity we aspirate >5 mm, with clear PET avidly also LN<5 mm are aspirated when technically feasible and when applicable EUSb approach was additionally used routinely. This has been changed in the manuscript line 155-156, as reviewer 2 also mentioned this. Routinely 3 passes are done, less if ROSE already confirm diagnosis, more if ROSE tells us to. Our EBUS protocol follows our earlier published studies on EBUS that also included ROSE by the same team: (MEDIAST Trial in J Clin Oncol 2023, 41, 3805-3815, doi:10.1200/JCO.22.01728; N1/Central Tumor EBUS: Chest 2015 Vol. 147 Issue 1 Pages 209-215; GRANULOMA Trial: JAMA 2013 Vol. 309 Issue 23 Pages 2457-64).
We added these topics for clarification to the limitations section of our revised manuscript, in line 448-450 in the Discussion.
-5-
Patients with ceCT performed >6 weeks prior to NB were excluded from the analysis. Please rationalize this exclusion criterion. It is reasonable to assume that the vast majority of patients included in the study had cross-sectional imaging that identified the target lung nodule. The authors should explain: (a) why a patient that had a ceCT >6 weeks ago and has (for example) a lung nodule with a hilar lymph node would be excluded from this study, (b) why 6 weeks (please provide a reference if available), and (c) was the 6-weeks criterion also applied to FDG PET studies.
This topic was also questioned in major-comment -2- by this reviewer and we would like to refer to our rebuttal in that section. To repeat (a) using this strict cut-off did not influence final outcome, but it makes comparison for other centers possible. (b) The 6 week cut-off follows our national guideline available at this link. And (c): yes, the 6 week cut-off was also applied to the PET-studies, see also line 128.
-6-
Line 164-166 – suggest flipping with the two sentences for better flow. Would be helpful to provide a summary of the diagnostic yield, diagnostic accuracy, and the lung nodule sampling range of pathologic diagnoses. What was the outcome of patients with nondiagnostic peripheral lung biopsy by NB? Along the same line, in Table 1 it is noted that in 21 cases the tumor histology is classified as “unknown” – how was the diagnosis of lung cancer made? This needs to be clarified in the manuscript.
Lines 180 and 182 (previously lines 164 and 166) are switched following your advice.
Information on diagnostic accuracy in this group has been added to line 106-107 in the Methods section, as this information was analyzed by Verhoeven et al. (2021).
We classified final outcome as “unknown” in cases where surgery was performed in their local hospitals and we did not receive detailed reporting from those patients e.g. only TNM-staging was available but not the specific pathology report. We have added a statement on this in line 124-125 to clarify.
-7-
Table 1- “combination of treatments” – please elaborate. Are neoadjuvant cases included here? Radio-chemotherapy? Also- chemotherapy is now frequently combined with immunotherapy. The separation between the two is unclear. Please provide a more detailed breakdown.
In response to the reviewer we have added more detail to Table 1. Neo-adjuvant treatment was offered in 10 cases only, this reflects the timeline of our study where neo-adjuvant treatment was not yet part of the routine practice as it is now. These cases were most likely patients that participated in clinical trials. Indeed, since these treatments are applied between EBUS and final surgery, this may affect outcome of the ypTNM staging.
With our term “Combinations of treatments” we grouped radio-chemotherapy (concurrent and sequential), chemo-immunotherapy, and chemo-radiotherapy followed by immunotherapy, all three or a combination of either treatment with local treatment. Since our manuscript focuses on EBUS performance for nodal detection in patients with early-stage lung cancer we believe that elaboration on this subgroup would deviate from the main research question and would only add to the complexity of Table 1.
-8-
Figure 1: please expand the greyed out box. The box title state PET or CT but there is no reference to how many patients were excluded due to lack of PET imaging. Also, patients with non-contrast enhanced CT were also excluded regardless of timeframe. This is not reflected in the box title.
Thank you for pointing out that this was not clear to you. Patients were firstly characterized based on PET-imaging. If no PET-imaging was available, we checked ceCT imaging for the presence of lymph nodes larger than 10 mm and only excluded patients based on availability of ceCT imaging within 6 weeks. In 212 cases no PET information was available < 6 weeks prior to navigation procedure (550 patients with a diagnostic NB, Figure 1) of which 338 had PET imaging < 6 weeks (Table 1). Of those 212 patients, 73 cases only had a non-contrast CT, 74 cases had a ceCT but older than 6 weeks, and then 65 patients had a ceCT < 6 weeks prior to NB and were included [table 1]. We have adjusted Figure 1 to clarify that patients with only a CT w/o contrast is across all time frames.
-9-
Based on Table 3, it appears that 27% (99/360) of LN samplings were non-diagnostic. There is no commentary about the outcome of these staging cases or how they were calculated in sections 3.2 or 3.3. Furthermore, this rate of non-diagnostic EBUS sampling appears to be higher than that reported on prior literature. The authors should comment on these two concerns.
Thank you for pointing this out. We would like to underline that our population is not to be compared to earlier publication on the general performance of EBUS in a population often selected on a much higher prevalence of mediastinal disease and prevalence of imaging suspected nodal involvement. To address this in more detail, we added more context in line 306-308 and line 490-492.
-10-
As in previous studies, occult nodal disease found on surgical resection has been attributable to lymph nodes not accessible by EBUS. Previous studies have found that EUS-FNA increases the detection of occult lymph node metastases. Please provide some rationale about why this should or should not be considered for similar patients undergoing navigational bronchoscopy, and the logistical or educational challenges that may ensue.
Thank you for pointing this out. As indicated above, we were part of the earlier randomized trials that have shown the added value of EUS and EUSb to routine EBUS. As such, when applicable EUSb approach was additionally used routinely. In this specific population in only one patient, station 8 (which could have been reached by EUSb-FNA) was involved but in this specific patient station 7 and 11R were also involved. We have added a statement on the addition of EUSb in our cohort in line 155-156.
-11-
The discussion section could benefit from a revision. Some of the discussions on how best to utilize PET-CT vs. CT with EBUS-TBNA could be more succinct. Instead of organizing the discussion by concept, would recommend using standard sections:
Restate research question/hypothesis
Summarize key findings
Interpret your findings
Compare to existing literature
Provide limitations
Provide implications
Provide future directions
Thank you for your advice. In response we have modified this section in the revised manuscript as suggested by the reviewer.
Minor comments:
Abstract:
Prevalence of lung cancer in patients referred for NB is high- based on what data? Several of the lung cancer screening studies, such as NLST, show otherwise.
Thank you for pointing this out. The high prevalence in this specific population is a result of the selection of cases with increased risk. In the Netherlands, screening for lung cancer has not (yet) been implemented and patients referred to our navigation bronchoscopy center had incidental nodules or were detected in the follow-up of other (often malignant) diseases. We do not refer to screening population, but to a population referred for biopsy. As such, looking specifically at the NB cohort, we have found a malignancy incidence of 76% (Verhoeven et al. (2021)) as found in the reference 5; we have updated these lines and references in line 65-75).
Introduction:
Lines 58-60 – the references provided do not support the statement. Please provide substantial references or revise the statement accordingly.
We have adjusted the statement to provide more context and have adjusted the references accordingly.
For lines 58-59 in the introduction: “The growing number of nodules...” recommend rephrasing.
We have adjusted the first paragraph of the introduction to better adhere to the purpose of the manuscript as recommended by the reviewer.
Lines 60-61 - the references provided do not support the statement. Please provide substantial references or revise the statement accordingly.
We have added a reference to include the higher number of navigation bronchoscopies following the rise in imaging that sees pulmonary nodules. However, Verhoeven et al. (2021) states the prevalence of lung cancer patients across our own NB cohort and therefore represents this point correctly.
For lines 60-61, recommend removing, “As a result,” and rephrase as “In addition, the prevalence of lung cancer...” Would also recommend providing numbers of lung cancer prevalence in patients getting navigational bronchoscopy from references 4 and 5.
We have added the prevalence of lung cancer as seen in our own cohort (76%) to this paragraph to add more context to the statement.
Line 62 – the term “effective” is vague. Please revise to provide a meaningful statement that truly reflects the role and efficacy of NB in the diagnosis of peripheral lung nodules (which is not as good as TTNB).
We agree that the term effective could be improved upon and have revised the first two sentences of this paragraph.
Line 70 – “All patients” - Based on what data? Please provide a supportive reference or revise.
We have removed ‘all’ and believe the argument is now still valid and conveys the correct message.
Line 74, “Although these criteria...” it is unclear what is meant by criteria, please clarify or rephrase.
We have replaced these criteria by guidelines on the performance of EBUS to improve on the information that is given here.
Line 79-80 – the deabbreviated FDG should be moved up to line 70
This was indeed missed and has been adjusted accordingly. We would like to thank the reviewer for their thorough reading of the manuscript and pointing this out.
Line 84: the use of the term “small” is misleading. The study included lesions both smaller and greater than 3 cm, which includes both nodules and masses. Please revise.
In line with above, we have removed the term small here and everywhere else in the manuscript where it was not appropriate, because indeed not all nodules of patients in the cohort could be considered small.
Methods:
Lines 103-105 – this exclusion criterion is unclear. Please redefine “uncertain benign outcome with insufficient or incomplete clinical follow-up” – this statement is unclear. The cited Delphi statement does not indicate that these cases should be excluded, but rather cannot be included in accuracy analyses. If that was the reason for exclusion, please state that clearly in the methodology.
We have added that these patient could not be included in the analysis since they could not be categorized, in line 114.
Line 109 – please redefine “aspect of the nodule”
We have changed aspect of the nodules to radiographic characteristics of the nodule and have also adjusted this in Table 1.
Line 112 – data is a plural term. Please correct.
We have adjusted the line to refer to data as a plural term.
Remove line 129 in Methods, it is a repeated sentence of line 130.
Thank you for pointing this out, we have removed it.
Results:
How many cases had ROSE available?
All cases had ROSE available, but this was indeed not clear from the manuscript. We have added this in line 147-148.
Line 160 – consider reporting median with range (instead of mean) to align with Table 1.
We have indeed made a typo here and reported the median as a mean. We have adjusted mean to median, since that is the correct descriptive method used here.
Line 188 – the authors should provide quantifiable details about the number of cases where EBUS was not performed.
There are unfortunately no numbers available on the reasons that the endoscopist refrained from performing an EBUS. All involved clinician were are highly experienced endoscopists in our center, but details were not quantifiable.
Lines 196-197 – recommend moving the statement regarding available surgical resection data earlier in this paragraph for more clarity.
We would like to thank the reviewer for this advice, and agree that moving the statement improves readability and also improves flow. We have adjusted the placement of this information to earlier in the paragraph.
Although briefly mentioned under sections 3.2 and 3.3, would interesting to include data on the number of neoadjuvant cases included in this study and whether that affected pre- and post-resection staging.
We would like to refer to our response above (major comment # 7). EBUS staging was always performed prior to any neo-adjuvant treatment, and could only affect ypTNM in surgical cases with lung cancer. In the present cohort no patients were (up- or) downstaged after surgery as can be seen in the Figure 2 and 3.
The results would benefit from adding the median number of lymph nodes sampled per patient in
Consider including data—if available—on median procedural time for navigational bronchoscopy and when combined with EBUS.
We added the number of sampled stations in the imaging positive and imaging negative groups in Table 1, with a median of 2 and 1 lymph node stations respectively.
Regarding the second point; while this would indeed be very interesting, this information is unfortunately not available.
Table 1, when EBUS-TBNA was performed. This is helpful for future studies.
We have adjusted the statement on reasoning behind performing TBNA in the Methods, line 153-156. The number of patients that received TBNA and that did not is stated in Table 1.
Table 1: consider changing “aspect” to “radiographic consistency”
We have adjusted this to radiographic characteristics in line 120 and in Table 1.
Table 1: nodules per patient: best to provide a summary of how many cases included 1 targeted lesion, 2 targeted lesions and so forth. Providing a mean or a median (unclear what is reported in the table) is not clear.
We agree with the reviewer and in the revised manuscript provide context to the complexity of the NB procedure, as suggested and describe patients with one, two or three nodules in Table 1.
Figure 2 – some numbers do not add up. For example: under “EBUS outcome of lymph nodes in lung cancer patients”, under cN0 there are 63+16 patients, but the n is indicated as 77. The authors should revisit the figure and revise the counts.
Thank you for pointing this out. We have corrected the typo’s revised two number in Figure 2 and now everything adds up correctly in the revised version.
Discussion:
Lines 352 and 281 are contradictory and need to be clarified. In line 352: “Our cohort of patients referred...” describes the patient population as “heavily preselected to exclude patients with a high risk of mediastinal [disease].” In line 281 of the Discussion, the cohort is described as an “unselected group of patients.”
While it is an unselected patient group in our navigation program, these patients have been referred for navigation bronchoscopy and as such are preselected. We therefore believe that both can be true, but have refrained from the term heavily, as this could create a false indication of the analyses performed in the study, in line 471.
Lines 352-354, consider rephrasing for ease of reading.
This line has been rephrased, as it contained the same statement twice.
In reference to lines 357-360 in the discussion, “This implies that...” it should also be stated that the study findings support the current ACCP and ESTS guidelines that mediastinal staging should be considered for FDG-avid lymph nodes regardless of lesion size.
Thank you for pointing this out. We have changed this in the introduction (line 98) and added that the findings of our study still support decision making as per the ACCP and ESTS guidelines in line 479-480, and do not follow the often-used clinical argumentation to support navigation bronchoscopy over TTNB (when this would be possible).
Round 2
Reviewer 3 Report
Comments and Suggestions for Authors
We thank the authors for comprehensively addressing our comments and revising the manuscript accordingly to the benefit of transparency and clarity.
Under comment #6 the authors note "Information on diagnostic accuracy in this group has been added to line 106-107 in the Methods section, as this information was analyzed by Verhoeven et al. (2021)." - does that mean that diagnostic yield/accuracy data from the current cohort was previously published by the group? If so, the authors should clearly state that in the manuscript.
Comments on the Quality of English LanguageThis manuscript would benefit from professional English editing.
Author Response
Reviewer:
We thank the authors for comprehensively addressing our comments and revising the manuscript accordingly to the benefit of transparency and clarity.
Under comment #6 the authors note "Information on diagnostic accuracy in this group has been added to line 106-107 in the Methods section, as this information was analyzed by Verhoeven et al. (2021)." - does that mean that diagnostic yield/accuracy data from the current cohort was previously published by the group? If so, the authors should clearly state that in the manuscript.
Response:
Dear reviewer, thank you for the fast reply to our response. We confirm as stated in the initial submission letter that none of the data used for this study have been published before.
In the discussion we had with you on the topic of diagnostic yield we mentioned one of our earlier publications as reference to support the number used, but that earlier manuscript (Verhoeven et al. (2021) - ref 5 in the current manuscript) used a comparable, much earlier cohort of patients in our center and addressed another topic. To avoid any misunderstanding we have added the words new cohort in the opening sentence of the Discussion in line 299.